# Effects of Dietary Glucose and Fructose on Copper, Iron, and Zinc Metabolism Parameters in Humans

**DOI:** 10.3390/nu12092581

**Published:** 2020-08-25

**Authors:** Nathaniel H. O. Harder, Bettina Hieronimus, Kimber L. Stanhope, Noreene M. Shibata, Vivien Lee, Marinelle V. Nunez, Nancy L. Keim, Andrew Bremer, Peter J. Havel, Marie C. Heffern, Valentina Medici

**Affiliations:** 1Department of Chemistry, University of California Davis, One Shields drive, Davis, CA 95616, USA; nhharder@ucdavis.edu; 2Department of Molecular Biosciences, School of Veterinary Medicine, University of California Davis, 2211 VM3B, Davis, CA 95616, USA; bettina.hieronimus@mri.bund.de (B.H.); klstanhope@ucdavis.edu (K.L.S.); vilee@ucdavis.edu (V.L.); mvnunez@ucdavis.edu (M.V.N.); pjhavel@ucdavis.edu (P.J.H.); 3Max Rubner-Institut, Institute of Child Nutrition, Haid-und-Neu-Strasse 9, 76131 Karlsruhe, Germany; 4Department of Internal Medicine, Division of Gastroenterology and Hepatology, University of California Davis, 4150 V Street, PSSB Suite 3500, Sacramento, CA 95817, USA; nshibata@ucdavis.edu; 5Department of Nutrition, University of California Davis, One Shields drive, Davis, CA 95616, USA; nancy.keim@usda.gov; 6United States Department of Agriculture, Western Human Nutrition Research Center, Davis, CA 95616, USA; 7Department of Pediatrics, School of Medicine, University of California Davis, 2516 Stockton Blvd, Ste 384, Sacramento, CA 95817, USA; andrew.bremer@nih.gov

**Keywords:** copper, ceruloplasmin, ferroxidase, sugar beverages, metabolism, lipid

## Abstract

Alterations of transition metal levels have been associated with obesity, hepatic steatosis, and metabolic syndrome in humans. Studies in animals indicate an association between dietary sugars and copper metabolism. Our group has conducted a study in which young adults consumed beverages sweetened with glucose, fructose, high fructose corn syrup (HFCS), or aspartame for two weeks and has reported that consumption of both fructose- and HFCS-sweetened beverages increased cardiovascular disease risk factors. Baseline and intervention serum samples from 107 participants of this study were measured for copper metabolism (copper, ceruloplasmin ferroxidase activity, ceruloplasmin protein), zinc levels, and iron metabolism (iron, ferritin, and transferrin) parameters. Fructose and/or glucose consumption were associated with decreased ceruloplasmin ferroxidase activity and serum copper and zinc concentrations. Ceruloplasmin protein levels did not change in response to intervention. The changes in copper concentrations were correlated with zinc, but not with iron. The decreases in copper, ceruloplasmin ferroxidase activity, ferritin, and transferrin were inversely associated with the increases in metabolic risk factors associated with sugar consumption, specifically, apolipoprotein CIII, triglycerides, or post-meal glucose, insulin, and lactate responses. These findings are the first evidence that consumption of sugar-sweetened beverages can alter clinical parameters of transition metal metabolism in healthy subjects.

## 1. Introduction

Copper, iron, and zinc are essential transition metals, participating as cofactors in the activity of numerous enzymes. They can contribute to crucial metabolic processes, including energy and lipid metabolism, one-carbon metabolism, electron transport chain, neuromodulation, and antioxidant mechanisms [1,2,3]. Given the fine limits existing between the physiological concentrations at which these metals are essential and their toxic pro-oxidant activities when present in excess, interconnected transporters and chaperones tightly regulate absorption and metabolism.

Copper is absorbed throughout the small intestine via copper transporter 1 (CTR1) [4]. In the portal circulation, a significant portion of copper is loosely bound to albumin and delivered to hepatocytes where specialized chaperones carry copper to its specific metabolic sites. In particular, copper is delivered to the transporter ATPase copper transporting beta (ATP7B) which is responsible for the excretion of copper via ceruloplasmin [5]. Ceruloplasmin is the main plasma copper carrier, believed to carry 40–70% of copper in plasma of healthy people. Its copper-bound form, known as holoceruloplasmin, carries an average of six copper atoms, but it can also circulate as its copper-free form, apoceruloplasmin [6]. Ceruloplasmin is essential to iron transport and metabolism through its ferroxidase activity, which catalyzes the conversion of Fe^2+^ to Fe^3+^. Ceruloplasmin plasma levels are of clinical relevance, mainly in the diagnosis of Wilson disease, a genetic condition caused by *ATP7B* pathogenic variants [7,8,9]. Changes in plasma copper levels and parameters of copper metabolism are connected to metabolic conditions, including obesity and fatty liver, and dietary factors [2,10]. However, the significance of copper excess or deficiency in metabolism and the methods to measure them are subjects of ongoing debate. Epidemiological data over the past few decades show declining mineral and trace element contents in the Western diet [11,12]. In obese patients, plasma copper levels are positively correlated with body mass index (BMI) [13]. Obesity is accompanied by increased plasma ceruloplasmin levels and increased copper concentration in visceral fat [14]. Conversely, patients with fatty liver present lower hepatic and plasma copper and lower plasma ceruloplasmin levels compared to healthy people and patients with other liver diseases [15]. In addition, hepatic copper concentration is inversely correlated with the severity of hepatic steatosis and features of metabolic syndrome [15]. The provision of a high fructose diet in rats caused worsening of fatty liver, particularly when associated with marginal copper deficiency [16].

However, copper is not the only transition metal playing a role in metabolism and the antioxidant response. Both iron and zinc have equally complex and important regulatory roles in the body, playing essential functions as co-factors in energy production and homeostasis. Misregulation of these metals has been linked to numerous diseases, from metabolic to neurodegenerative diseases [17,18,19]. Iron, similarly to copper, can undergo redox active chemistry, having disastrous effects on cells and the body if not properly regulated [20,21,22]. Ferritin (an iron storage protein) and transferrin (an iron chaperone) are traditionally used alongside iron concentrations as biomarkers for serum iron status [23,24]. Zinc, on the other hand, is not redox active yet imparts significant function to a large number of proteins in both structural and enzyme active sites [25]. These metals have been shown to play significant roles in lipid metabolism. For instance, PPARγ, which regulates lipid metabolism, utilizes zinc fingers for DNA binding and is downregulated in zinc deficient conditions, while intracellular catalytic iron is believed to facilitate lipid peroxidation [26,27,28]. As such, understanding how these metals and their markers are affected by environmental factors, such as dietary sugar intake, is likely to prove important in the diagnosis or treatment of metabolic diseases.

Human and in vivo data in the metal micronutrient field are lacking. Clinical parameters assessing systemic metal status and metabolism and flow between organs are limited, likely due to discrepancies in the available data and the difficulties in their interpretation. In the case of copper, most clinical laboratories provide plasma ceruloplasmin protein levels assessed by immunologic assay; however, this parameter does not differentiate between apoceruloplasmin and holoceruloplasmin [29,30,31]. Ceruloplasmin (enzymatic) ferroxidase activity could aid in the direct quantification and speciation of copper in plasma, but it is not normally provided in clinical practice. Other parameters, including radioactive copper incorporation and exchangeable copper, appear promising and could potentially enable a better assessment of copper status, but remain unvalidated [32,33]. To study the effects of dietary components such as sugar consumption on transition metal levels and the resulting metabolic changes, improved clinical parameters for assessing these metal levels are necessary.

In the current study, we analyzed serum copper, iron, and zinc metabolism parameters and their relation to lipid metabolism in healthy, younger (18–40 years) male and female subjects who consumed beverages sweetened with glucose, fructose, high fructose corn syrup (HFCS), or aspartame for two weeks. We provide evidence that intake of sugar-sweetened beverages results in changes of copper, iron, and zinc status and metabolism in association with markers of lipid metabolism.

## 2. Materials and Methods

### 2.1. Human Subjects and Study Design

Subjects in this study are a subgroup from a NIH-funded, parallel-arm, double-blinded diet-intervention study. Subject characteristics and detailed study design were previously reported [34,35]. In brief, the study occurred in three phases: (I) 3.5-day inpatient baseline; (II) 12-day outpatient intervention; and (III) 3.5-day inpatient intervention. Subjects resided at the UC Davis Clinical and Translational Science Center Clinical Research Center (CCRC) during inpatient periods. Each experimental group was matched according to sex, BMI, fasting triglyceride (TG), cholesterol, high-density lipoprotein (HDL), and insulin. This study reports results from 107 of the 187 original participants consuming beverages containing aspartame (*n* = 23), glucose (*n* = 28), fructose (*n* = 28), and HFCS composed of 45% glucose and 55% fructose (*n* = 28). For 5 weeks prior to the study, subjects were asked to restrict daily intake of sugar-containing beverages to one 8-oz serving of fruit juice and stop consuming all vitamin and mineral dietary or herbal supplements.

Informed written consent was obtained from each subject and the study protocol conformed to the ethical guidelines of the 1975 Declaration of Helsinki as reflected in a priori approval by the Institutional Review Board at the University of California, Davis; IRB #214709 (19 August 2018) and #1332484 (5 December 2018).

### 2.2. Diet

During the 12-day outpatient phase, subjects were provided with and instructed to drink three servings of sweetened beverage per day (one per meal), consume their normal diet, and abstain from other sugar-containing beverages, including fruit juice. The sugar-sweetened beverages contained glucose (STALEYDEX^®^ crystalline dextrose, Tate and Lyle, Hoffman Estates, IL, USA), fructose (KRYSTAR^®^ crystalline fructose, Tate and Lyle, Hoffman Estates, IL, USA), or HFCS (ISOSWEET^®^ 5500, Tate and Lyle, Hoffman Estates, IL, USA). These beverages were formulated as 15% sugar in water (weight/weight) and flavored with unsweetened Kool-Aid^®^ drink mix (Kraft Foods, Northfield, IL, USA). The three daily servings of sugar-sweetened beverage in total provided 25% of daily energy requirement, equivalent to approximately 4.5 12-ounce cans of a standard soft drink per day. Daily energy requirements were calculated by the Mifflin equation with adjustments of 1.3 for activity on the inpatient 24-h serial blood collection days and 1.5 for the other days [36]. The control beverages were sweetened with aspartame (Market Pantry™ sugar-free drink mix, Target Brands Inc., Minneapolis, MN, USA) and were provided as 3 daily servings comparable in quantity to the sugar-sweetened beverages. Riboflavin was added as a compliance biomarker to the sweetened beverages and measured by fluorimetry in urine collected from the subjects during both the inpatient and outpatient phases [37]. Urinary riboflavin levels did not differ between groups, and levels during the outpatient phase were not different than levels during the inpatient phase when consumption was monitored [34,35]. Subjects were instructed to abstain from alcohol the day before check-in to the CCRC.

### 2.3. Serum Metals and Ceruloplasmin Ferroxidase Activity

Fasting serum (30-min clotting time) was collected between 07:00 and 08:00 following subject check-in during both the baseline and intervention inpatient phase. Serum and sweetened beverage aliquots were digested in 1N Ultrex II nitric acid (Avantor JT Baker, Radnor Township, PA, USA) at 4 °C for 24 h then centrifuged at 3000 rpm/4 °C for 12 min in a Sorvall Legend X1R centrifuge (Thermo Fisher Scientific, Waltham, MA, USA) and supernatant collected. Iron, copper, and zinc levels were measured by inductively coupled plasma mass spectrometry at the UC Davis Interdisciplinary Center for Plasma Mass Spectrometry. Sweetened beverage samples were analyzed separately to determine if metal content in the Kool-aid drink mixes were significantly different from one another; no significant differences were found (Appendix A). Ceruloplasmin ferroxidase activity was measured with a colorimetric assay (EIACPLC, Invitrogen, Carlsbad, CA, USA). According to the manufacturer’s protocols, samples were diluted 1:50 in the commercial assay buffer and a colorimetric ceruloplasmin substrate was added. Plates were incubated at 30 °C for 60 min and absorbances were read at 560 nm using a Synergy H1 microplate reader (Bio Tek, Winooski, VT, USA).

### 2.4. Serum Protein Levels

Serum protein levels were measured using commercially available ELISAs according to the manufacturers’ protocols. Samples were diluted 1:400,000 for ceruloplasmin (EC4201-1, Assaypro, St. Charles, MO, USA); 1:250,000 for transferrin (ab187391, Abcam, Cambridge, MA, USA); and 1:10 for ferritin (ab200018, Abcam, Cambridge, MA, USA). Samples and standards were run in duplicate and their absorbance was measured using a Synergy H1 microplate reader (Bio Tek, Winooski, VT, USA).

### 2.5. Plasma Lipid and Glucose Parameters

As previously reported [34,35], 24-h serial plasma samples were collected every 30 or 60 min starting at 08:00 during the third day of each inpatient phase and analyzed for TG, lipoproteins, insulin, and glucose. In the current study these data were utilized for the purpose of novel correlation analyses with metal and metal biomarker parameters. Plasma TG concentrations were measured at all time points and calculated for mean 24-h concentration and for total 24-h area under the curve (AUC) by the trapezoidal method. TG, apolipoprotein CIII (apoCIII), and lactate concentrations were measured with a Polychem Chemistry Analyzer (PolyMedCo Inc., Cortland Manor, New York, USA) with reagents from MedTest DX (Canton, MI, USA). Glucose was measured with an automated glucose analyzer (YSI, Inc., Yellow Springs, OH, USA) and insulin by radioimmunoassay (Millipore, St. Charles, MO). Changes in glucose and insulin amplitude were calculated as post-meal zenith minus pre-meal nadir concentration for each meal and averaged for the 3 meals.

### 2.6. Statistical Analyses

Metal concentrations in the sweetened beverages and baseline variables were analyzed by one-way ANOVA.

Each metal outcome was analyzed for the effect of beverage group in a MIXED procedure repeated measures (time) analysis of covariance (ANCOVA) using SAS 9.4 (Cary, NC, USA). The model included adjustments for sex, BMI, time, and beverage x time, with Tukey’s post-test for beverage x time. Identical repeated measures ANCOVAs that included adjustment for the change in BMI were also conducted. The percentage change of each outcome was analyzed by the general linear model ANCOVA; the beverage interventions were included as its proportional contents of monosaccharides (fructose and glucose) as separate variables. Therefore, fructose was input as 1 for fructose and 0 for glucose, glucose as 0 for fructose and 1 for glucose, HFCS as 0.55 for fructose and 0.45 for glucose, and aspartame as 0 for both fructose and glucose. The model was adjusted for outcome at baseline, BMI, and sex. Covariates that decreased the sensitivity of the model were removed. Outcomes significantly affected by BMI were further assessed by two-tailed unpaired *t*-test (Microsoft Excel; Microsoft, Redmond, WA, USA) with subjects separated into two cohorts by BMI < or ≥ 25 kg/m^2^.

Partial Pearson correlations adjusted for BMI and sex (SAS 9.4) were used to assess the relationships between: (1) metal markers at baseline; (2) changes in metal markers; and (3) changes in metal markers and metabolic markers. Significant relationships between changes in metal markers and metabolic markers were further tested by partial Pearson correlations adjusted for fructose, glucose, BMI, and sex.

*p*-values < 0.05 were considered significant.

## 3. Results

### 3.1. Baseline Measures

There were no significant differences in baseline characteristics between the four intervention groups (Table 1) or in the copper, iron, and zinc content of the study beverages (Appendix A).

### 3.2. Effects of Glucose, Fructose, HFCS, and Aspartame on Metals and Metal Metabolism Markers

Table 2 shows the absolute values of the seven metal-associated outcomes (three metals and four metal metabolism markers) at baseline and after intervention. Ceruloplasmin concentration was not affected by the consumption of the sweetened beverages. The serum concentrations of all other metal-associated outcomes decreased over time; however, there were no significant effects of beverage group or beverage x time (Table 3). Subjects consuming HFCS and glucose exhibited the most marked decreases in copper concentrations (HFCS: −83.9 ± 25.3, 2 weeks vs. 0 weeks, Tukey’s post-test adjusted *p*-value: 0.0019; glucose: −77.6 ± 13.8 ppb; adjusted *p*-value = 0.0069). Ceruloplasmin ferroxidase activity was also decreased following HFCS or glucose consumption; however, when adjusted for multiple comparisons, the decreases were not significant (HFCS: −2.0 ± 0.53 U/mL, unadjusted *p*-value: 0.0049; adjusted *p* = 0.089; glucose: −2.1 ± 0.9 U/mL, unadjusted *p*-value: 0.0029, adjusted *p*-value: 0.056). Subjects consuming HFCS or glucose also exhibited decreased zinc concentrations (HFCS: −93.3 ± 29.3 ppb; Tukey’s post-test unadjusted *p*-value: 0.0006; adjusted *p*-value: 0.014; glucose: −80.0 ± 25.2 ppb; Tukey’s post-test unadjusted *p*-value: 0.0032; adjusted *p*-value: 0.060, 2 weeks versus 0 weeks). Adjustment of the repeated measures ANCOVAs for the change in BMI tended to strengthen rather than attenuate these results and did not reveal any additional effect of the beverage interventions.

Sex was a significant factor in the model (Table 3). However, the differences between male and female in copper, ceruloplasmin ferroxidase activity, zinc, iron, and ferritin levels were specific to baseline and not due to an effect of intervention.

There were significant relationships between BMI and both iron and copper (Table 3). To further investigate these relationships, subjects were stratified by BMI (BMI < 25 and BMI ≥ 25), and iron and copper concentrations were compared at both baseline and intervention (Table 4). Serum copper concentrations tended to be higher and iron concentrations lower at baseline in subjects with BMI ≥ 25 compared to those with BMI < 25. After intervention, copper was significantly higher in subjects with BMI ≥ 25, but a BMI-influenced difference in iron was no longer apparent.

To further investigate the changes over time, we analyzed each outcome in a model that assessed the specific effects of glucose and fructose. Copper and ceruloplasmin ferroxidase activity were significantly influenced by the effects of fructose and/or glucose (Figure 1). Glucose, consumed at 25% of energy requirement, accounted for −7.06% ± 3.26% of the change in copper (*p* = 0.033) and −9.23% ± 3.46% of the change in ceruloplasmin ferroxidase activity (*p* = 0.009). Fructose, consumed at 25% of energy requirement, accounted for −7.26% ± 3.4% of the change in ceruloplasmin ferroxidase activity (*p* = 0.035). Glucose consumption also contributed to the decrease in serum zinc concentrations (−8.41% ± 4.1%, *p* = 0.045) (Figure 1). Neither glucose nor fructose consumption contributed to the significant decreases in iron status markers observed during the two-week intervention (data not shown).

### 3.3. Correlations of Baseline Serum Metal Concentrations with Baseline Metal Metabolic Markers

Partial Pearson correlations of baseline metals and their metabolism markers were adjusted for BMI and sex (Table 5). This analysis of baseline metal markers presents some interesting correlations, with copper positively correlated to ceruloplasmin ferroxidase activity (r = 0.520, *p* < 0.001), but not ceruloplasmin concentration (r = 0.125, *p* = 0.204). In addition to this lack of significance with copper, ceruloplasmin concentration is negatively correlated to its ferroxidase activity (r = −0.452, *p* < 0.001). Zinc was also found to be negatively correlated with ceruloplasmin concentration (r = −0.208, *p* = 0.033). Iron concentrations were positively correlated to both transferrin (r = 0.350, *p* < 0.001) and ferritin (r = 0.241, *p* = 0.014).

### 3.4. Correlations of Changes in Serum Metal Concentrations with Changes in Metal Metabolism Markers

Correlations of the changes from week 0 to week 2, adjusted for BMI and sex, highlighted different interactions between metal markers than observed with baseline correlations (Table 6). ΔCopper was strongly positively correlated with Δzinc (r = 0.359, *p* < 0.001) and Δceruloplasmin ferroxidase activity (r = 0.572, *p* < 0.001) along with a weaker positive correlation with Δceruloplasmin concentration (r = 0.235, *p* = 0.016). ΔIron concentrations were strongly correlated with only Δtransferrin (r = 0.445, *p* < 0.001). These significant relationships were not attenuated in additional partial correlation analyses that included adjustment for fructose, glucose, BMI, and sex (data not shown).

### 3.5. Correlations of Changes in Serum Metal Concentrations with Changes in Metabolism Markers

Partial correlations, with adjustments for BMI and sex, between changes from week 0 to week 2 of metal markers, and changes in selected metabolic markers showed negative correlations (Table 7). ΔCopper was negatively correlated with Δfasting levels of apoCIII (r = −0.25, *p* = 0.011). ΔCeruloplasmin ferroxidase activity was negatively correlated with the Δamplitude of glucose (r = −0.260, *p* = 0.007), insulin (r = −0.270, *p*= 0.007), and lactate (r = −0.300, *p* = 0.003). ΔIron metabolism was also related to changes in metabolic markers. ΔTransferrin concentration was negatively correlated with Δpostprandial apoCIII (r = −0.200, *p* = 0.046) and Δferritin was negatively correlated with Δpostprandial TG levels (r = −0.220, *p* = 0.025), ΔTG AUC (r = −0.230, *p* = 0.021), Δfasting apoCIII (r = −0.290, *p* = 0.003), Δpostprandial apoCIII (r = −0.290, *p* = 0.003), and Δlactate amplitude (r = −0.21, *p* = 0.039). These significant inverse relationships were not attenuated in additional partial correlation analyses that included adjustment for fructose, glucose, BMI, and sex (data not shown).

## 4. Discussion

The current study provides insights on the relationship between the consumption of sweetened beverages in the diet and metal micronutrient regulation in humans. Our data propose connections between diet composition; copper, iron, and zinc metabolism; and metabolic biomarkers in healthy subjects. We found consumption of both glucose and fructose contributed to significant decreases in ceruloplasmin ferroxidase activity but had no effect on ceruloplasmin concentrations. We also found consumption of glucose, but not fructose, contributed to decreases in serum copper and zinc concentrations.

The notable changes in ceruloplasmin ferroxidase activity and copper concentrations in the serum of subjects consuming sugar-sweetened beverages suggest a connection between sugar ingestion and copper metabolism. The role of copper in human health is poorly understood, with a majority of research investigating copper in patients with obesity. Our data point to healthy adults having increased serum copper levels with higher BMI. The association between serum copper concentration and adiposity status in healthy individuals was also noted by Olusi et al., who reported a positive association between copper and leptin in healthy adults [38]. Yang et al. reported that an increase in copper and ceruloplasmin serum concentrations was positively associated with BMI [14]. Additional studies have shown increased copper export along with decreased hepatic copper concentration in non-alcoholic fatty liver disease (NAFLD) and a relationship between elevated serum copper and liver cirrhosis or hepatocellular carcinoma [10,39,40]. While these studies relate elevated serum copper to obesity and associated conditions, our data suggest consumption of sugar-sweetened beverages, especially glucose containing beverages, by healthy subjects, decreases serum copper and ceruloplasmin ferroxidase activity. Similarly to our findings, Aigner et al. demonstrated patients with NAFLD present a decrease in both hepatic copper concentration and serum ceruloplasmin ferroxidase activity [41]. Previous in vivo studies indicate involved pathogenic factors may include reduced duodenal copper absorption due to inhibited transcription of *Ctr1*, concomitant hepatic iron accumulation, or alterations in gut microbiota composition possibly related to increased abundance of Firmicutes and reduced Akkermansia [16,41,42]. Other proposed mechanisms involve copper regulation of cyclic-AMP-dependent lipolysis or mitochondrial dysfunction associated with impaired cupro-enzymatic activity [43,44,45]. These mechanisms may be contributing factors to effects observed but not studied in our work.

To the best of our knowledge, this is the first study of the effects of glucose or HFCS on copper and its markers in healthy subjects. Previous studies have focused on fructose and indicated fructose may impact copper metabolism in a unique way [46,47]. In rats, fructose-rich diets and copper-deficient diets showed exacerbations of NAFLD-like pathology, suggesting a potential crosstalk between copper metabolism and fructose [46]. Recent studies have further corroborated this finding and revealed sex-specific differences, finding that a diet of 30% caloric intake from fructose altered copper only in males [48]. Our results suggest glucose consumption may also have a negative impact on copper metabolism and this impact may be more marked than that of fructose. More studies of the effects of glucose, HFCS, and fructose on copper metabolism are needed.

We found changes in copper were negatively associated with changes in fasting apoCIII, a lipoprotein implicated as a cardiovascular disease risk factor [49], and ceruloplasmin ferroxidase activity was negatively associated with increases in post-meal glucose, insulin, and lactate responses. These inverse associations, which were independent of the effects of fructose and glucose, suggest the possibility that adequate copper and ceruloplasmin ferroxidase activity is linked to fuel utilization and may have protective metabolic effects.

Traditionally it is thought that iron is linked to copper through the activity of ceruloplasmin, however, our data suggest that zinc may also be involved [50]. Serum zinc levels follow a pattern similar to copper, with glucose consumption contributing to decreased zinc concentrations. We also observed a strong positive association between the changes in zinc and copper independent of the effects of glucose and fructose. Baseline serum zinc concentrations were negatively correlated with ceruloplasmin concentration. A connection between sugar consumption and zinc has also been recently reported wherein rats consuming high fructose diets exhibited decreases in hepatic zinc concentrations [48]. These data strengthen the connection between copper and zinc, but also indicate that the interplay between them, especially in the context of dietary interventions, has yet to be fully understood [51,52,53].

While we failed to observe any significant difference between beverage groups, or significant effects of fructose or glucose, on the decreases in iron and iron markers, we observed some interesting relationships. Analysis of the change from baseline to intervention of iron and ferritin by repeated measures ANCOVA reveals that serum iron was significantly decreased in all four beverage groups. Ferritin, however, was significantly decreased in subjects consuming sugar-containing beverage, but not the subjects consuming aspartame. Literature is sparse regarding connections between serum iron and aspartame. As the aspartame-induced decrease in serum iron concentrations was not confirmed by an aspartame-induced decrease in ferritin, additional studies are required to determine whether aspartame consumption affects iron levels. We observed that subjects with BMI < 25 had higher serum iron concentrations at baseline than subjects with BMI ≥ 25. This difference requires further study as it was not observed at the end of the two-week intervention. However, the baseline relationship supports data reporting decreased serum iron concentration is associated with increased BMI [54]. Our data support the findings that NAFLD induces increased hepatic iron concentrations and increased hepatic hepcidin expression, which would result in decreased serum iron levels [41]. This relationship between hepcidin and diet merits additional research given the role of this peptide in the innate immune system [55]. Studying the interplay between diet, inflammatory cytokines, and transition metal regulation would add another piece to the puzzle of transition metals in biology.

We also observed that the changes in ferritin were negatively associated with the changes in postprandial TG and the changes in fasting and postprandial apoCIII. These effects, which were independent of fructose and glucose consumption, are contradicted by a recent observational study showing fasting ferritin levels were positively associated with TG and lipoproteins [56]. This discrepancy may be due to the much higher ferritin concentrations exhibited by the subjects studied by Zhou et al. They were divided into quartiles based on ferritin concentrations; only three of the subjects in our study met the criteria of the second-highest quartile and none fit in the highest quartile.

One important observation of the present study concerns the relationship between serum copper concentration, ceruloplasmin levels, and copper-dependent ceruloplasmin ferroxidase activity. Laboratory tests for ceruloplasmin levels are widely used as a proxy for copper concentration in the serum [29,57]. While serum ceruloplasmin has been a key test for diagnosing Wilson disease, our findings suggest ceruloplasmin ferroxidase activity might be a more accurate representation for copper status given the observed strong correlation of copper concentration with ceruloplasmin ferroxidase activity and a weaker correlation with ceruloplasmin concentration. The differences between activity and concentration are even more apparent when considering the effects of the intervention. The change in serum copper was positively correlated with ceruloplasmin ferroxidase activity, yet negatively correlated with ceruloplasmin concentration. This accentuates the differences between the activity and concentration of a protein and the need for additional studies untangling how these parameters relate to one another and to copper metabolism.

Our study is limited by the post hoc nature of the analysis. Consumption of metal micronutrients were not controlled during the outpatient period of the study and the subjects were instructed to consume their normal diets. Thus, both metal consumption and bioavailability could have been altered by variables for which this study did not control. In addition to variabilities in the subjects’ normal diets, it is possible that the added calories from the beverages could have displaced the consumption of metal-rich foods or nutrients that enhance metal bioavailability. It is also feasible that consumption of sweetened beverages affected metal absorption. Future studies may be informed by existing reports on the impacts of sugar intake on metal bioavailability [58,59]. In addition, given the assessment of metal markers was not initially planned, the study might also have been under-powered to assess certain effects of sugar consumption on metal metabolism. Future studies are warranted that control for metal micronutrient intake and being sufficiently powered to validate the trends observed in this study. Nevertheless, the short-term two-week intervention was sufficient to cause significant sugar-dependent changes in certain serum metal markers, offering support for a connection between sugar metabolism and metal homeostasis.

## 5. Conclusions

Our results suggest sugar consumption affects copper and zinc metabolism parameters within two weeks. Alterations in transition metal metabolism, in turn, can contribute to aberrant fuel utilization and potentially worsen the course of metabolic-related diseases. Glucose consumption may possibly have a more marked impact on transition metal metabolism than fructose consumption. Ceruloplasmin ferroxidase activity appears to correlate significantly with copper serum levels and may have broader applications in clinical practice than ceruloplasmin levels.

Our findings are the first evidence that consumption of sugar-sweetened beverages can alter clinical parameters of transition metal metabolism in healthy subjects. As sugar consumption has become entrenched in the Western diet, further study of the relationships between transition metal metabolism and glucose, fructose, HFCS, and also sucrose is warranted. Our findings represent the proverbial tip of the iceberg toward understanding the physiological impact of microminerals in the modern Western diet and highlight the need to further probe the clinical relationship between metal status and metabolic health.

## Figures and Tables

**Figure 1 nutrients-12-02581-f001:**
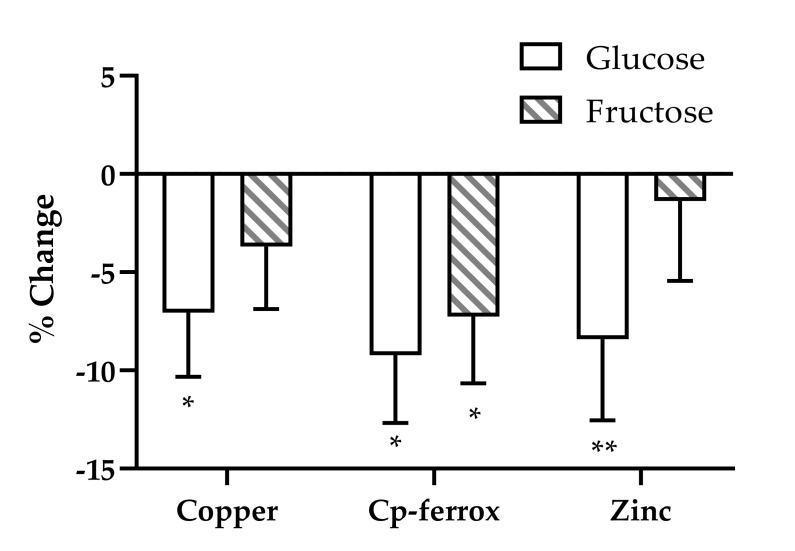
Estimates of the effects of fructose and glucose on the percentage changes of serum copper, Cp-ferrox, and zinc in subjects consuming beverages containing fructose, glucose, HFCS, or aspartame for 2 weeks. General linear model ANCOVA with beverage interventions described as the proportional contents of fructose and glucose as separate variables. Model included adjustments for BMI and sex. * *p* < 0.05, ** *p* < 0.01, effect of glucose or fructose.

**Table 1 nutrients-12-02581-t001:** Description of subjects at baseline by sugar consumed. Means ± SEMs are shown. BMI, body mass index; HFCS, high fructose corn syrup.

	Glucose *n* = 28	Fructose *n* = 28	HFCS *n* = 28	Aspartame *n* = 23	Total *n* = 107
BMI	25.4 ± 0.7	25.8 ± 0.7	24.9 ± 0.8	24.8 ± 0.7	25.3 ± 0.4
Age	26.8 ± 1.2	26.0 ± 1.1	26.8 ± 1.2	25.4 ± 1.3	26.3 ± 1.6
Sex	15 M/13 F	15 M/13 F	15 M/13 F	11 M/12 F	56 M/51 F
% M/F	53.6/46.4	53.6/46.4	53.6/46.4	47.9/53.1	52.3/47.7

**Table 2 nutrients-12-02581-t002:** Serum metals and metal metabolism markers during consumption of aspartame, glucose, fructose, or HFCS-sweetened beverages at week 0 and week 2. Means ± SEMs are shown. Cp, ceruloplasmin; Cp-ferrox, ceruloplasmin ferroxidase activity.

	Aspartame		Glucose		Fructose		HFCS	
	Week 0	Week 2	Week 0	Week 2	Week 0	Week 2	Week 0	Week 2
**Cp** **(µg/mL)**	773 ± 125	1095 ± 287	920 ± 157	815 ± 125	1098 ± 258	805 ± 198	885 ± 146	984 ± 156
**Cp-ferrox** **(U/mL)**	26.8 ± 2.5	26.8 ±2.3	28.2 ± 2.2	26.1 ± 1.8	28.8 ± 1.5	27.7 ± 1.5	29.5 ± 2.3	27.5 ± 2.1
**Transferrin** **(mg/dL)**	93.6 ± 6.6	84.87 ± 6.5	104.2 ± 4.4	104.4 ± 6.2	101.2 ± 5.1	94.0 ± 5.8	109.2 ± 5.7	94.0 ± 5.4
**Ferritin** **(ng/mL)**	37.7 ± 10.2	28.9 ± 7.54	48.7 ± 7.9	36.8 ± 7.1	45.6 ± 8.4	28.7 ± 5.7	41.9 ± 6.0	31.1 ± 5.3
**Copper** **(ppb)**	912 ± 49	881 ± 41	976 ± 55	869 ± 43	953 ± 34	898 ± 32	1019 ± 55	935 ± 49
**Iron** **(ppb)**	1029 ± 77	726 ± 68	920 ± 82	825 ± 78	1037 ± 87	796 ± 53	1007 ± 74	750 ± 59
**Zinc** **(ppb)**	975 ± 37	957 ± 30	1012 ± 24	932 ± 33	983 ± 24	969 ± 24	991 ± 33	897 ± 22

**Table 3 nutrients-12-02581-t003:** *p*-values for the MIXED procedure repeated measures (time) ANCOVA testing each outcome for the effects of beverage, sex, BMI, time, and beverage × time. Significant values are in bold.

	Beverage	Sex	BMI	Time	Beverage × time
	*p*-value	*p*-value	*p*-value	*p*-value	*p*-value
**Cp**	0.972	0.615	0.289	0.945	0.087
**Cp-ferrox**	0.873	**0.005**	0.255	**<0.001**	0.161
**Copper**	0.203	**<0.001**	**<0.001**	**<0.001**	0.331
**Zinc**	0.737	**<0.001**	0.558	**<0.001**	0.084
**Iron**	0.916	**0.001**	**0.039**	**<0.001**	0.442
**Ferritin**	0.878	**<0.001**	0.177	**<0.001**	0.284
**Transferrin**	0.104	0.355	0.590	**0.003**	0.209

**Table 4 nutrients-12-02581-t004:** Copper and iron concentrations in subjects divided by BMI into normal weight (<25 kg/m^2^) or overweight and obese (≥25 kg/m^2^) groups. An unpaired, two-tailed *t*-test was used to analyze significance between BMI groups. Mean ± SEM are shown.

		**BMI < 25** **(*n* = 51)**	**BMI ≥ 25** **(*n* = 56)**	***p*-Value**
**Copper**	**Baseline**	918 ± 33	995 ± 30	0.08
	**Intervention**	847 ± 29	936 ± 29	**0.03**
**Iron**	**Baseline**	1107 ± 64	899 ± 47	**0.008**
	**Intervention**	779 ± 53	776 ± 40	0.97

**Table 5 nutrients-12-02581-t005:** Partial correlation between baseline metal and metal metabolism markers adjusted for BMI and sex. *p* < 0.05 are shown in bold with positive associations (r > 0) shown in blue, and negative associations (r < 0) shown in red.

		Copper	Iron	Zinc	Cp-ferrox	Cp	Transferrin	Ferritin
Copper	r		−0.131	−0.082	0.520	0.125	0.014	−0.237
	*p*		0.184	0.411	**<0.001**	0.204	0.890	0.016
Iron	r	−0.131		0.048	−0.098	0.016	0.350	0.241
	*p*	0.184		0.628	0.321	0.872	**<0.001**	**0.014**
Zinc	r	−0.0815	0.048		0.132	−0.208	0.169	−0.088
	*p*	0.411	0.628		0.178	**0.033**	0.087	0.376
Cp-ferrox	r	0.520	−0.098	0.132		−0.452	0.028	−0.153
	*p*	**<0.001**	0.321	0.178		**<0.001**	0.778	0.119
Cp	r	0.125	0.016	−0.208	−0.452		−0.032	−0.014
	*p*	0.204	0.872	**0.033**	**<0.001**		0.746	0.889
Transferrin	r	0.014	0.350	0.169	0.028	−0.032		0.103
	*p*	0.890	**<0.001**	0.087	0.778	0.746		0.295
Ferritin	r	−0.237	0.241	−0.088	−0.153	−0.014	0.103	
	*p*	0.016	**0.014**	0.376	0.119	0.889	0.295	

**Table 6 nutrients-12-02581-t006:** Partial correlation between changes from week 0 to week 2 in metal and metal metabolism markers adjusted for BMI and sex. *p* < 0.05 are shown in bold with positive associations (r > 0) shown in blue.

		Copper	Iron	Zinc	Cp-ferrox	Cp	Transferrin	Ferritin
Copper	r		0.114	0.358	0.572	0.235	−0.046	0.069
	*p*		0.251	**<0.001**	**<0.001**	**0.016**	0.642	0.492
Iron	r	0.114		0.204	0.040	−0.032	0.445	0.049
	*p*	0.251		0.036	0.687	0.745	**<0.001**	0.624
Zinc	r	0.358	0.204		0.309	0.173	0.191	−0.091
	*p*	**<0.001**	0.036		0.687	0.078	0.052	0.360
Cp-ferrox	r	0.572	0.040	0.309		0.187	0.093	0.041
	*p*	**<0.001**	0.687	0.687		0.056	0.346	0.676
Cp	r	0.235	−0.032	0.173	0.187		0.017	0.143
	*p*	**0.016**	0.745	0.078	0.056		0.864	0.147
Transferrin	r	−0.046	0.445	0.191	0.093	0.017		0.045
	*p*	0.642	**<0.001**	0.052	0.346	0.864		0.648
Ferritin	r	0.069	0.049	−0.091	0.041	0.143	0.045	
	*p*	0.492	0.624	0.360	0.676	0.147	0.648	

**Table 7 nutrients-12-02581-t007:** Partial correlation between changes from week 0 to week 2 in metal and metabolism markers adjusted for fructose, glucose, BMI, and sex. *p* < 0.05 are shown in bold with r < 0 shown in red. TG AUC, triglyceride area under the curve; apoCIII, apolipoprotein CIII.

		PostprandialTG	TG AUC	Fasting apoCIII	Postprandial apoCIII	Glucose Amplitude	Insulin Amplitude	Lactate Amplitude
Copper	r	−0.001	−0.069	−0.250	−0.053	0.028	−0.116	−0.026
	*p*	0.994	0.504	**0.011**	0.606	0.788	0.261	0.800
Iron	r	0.029	−0.059	−0.165	−0.142	0.066	0.069	0.074
	*p*	0.777	0.565	0.109	0.169	0.525	0.504	0.473
Zinc	r	0.010	−0.061	0.023	−0.036	0.059	0.080	0.150
	*p*	0.922	0.555	0.823	0.728	0.566	0.436	0.144
Cp-ferrox	r	0.010	0.095	−0.089	0.0172	−0.260	−0.270	−0.300
	*p*	0.334	0.359	0.389	0.868	**0.007**	**0.007**	**0.003**
Cp	r	−0.056	−0.060	0.050	0.005	0.085	−0.077	−0.121
	*p*	0.586	0.559	0.630	0.959	0.411	0.457	0.240
Transferrin	r	−0.047	−0.069	−0.078	−0.200	−0.104	−0.059	0.007
	*p*	0.651	0.507	0.451	**0.046**	0.312	0.570	0.947
Ferritin	r	−0.220	−0.230	−0.290	−0.290	−0.210	0.096	−0.21
	*p*	**0.025**	**0.021**	**0.003**	**0.003**	0.934	0.351	**0.039**

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
