# Peer review of "Effects of Dietary Glucose and Fructose on Copper, Iron, and Zinc Metabolism Parameters in Humans"

_nutrients, 2020, doi:10.3390/nu12092581_

Round 1
Reviewer 1 Report
Overall, this is an interesting post-hoc analysis from a diet intervention study that helps the field move forward from animal models. The introduction provides an excellent summary of the current questions. The study design is sound and documented in prior work. The authors acknowledge the limitations of study and provide a framework to move forward in this important research area. There are some relatively minor comments that could be considered in revising the work as follows:
It is interesting that aspartame seemed to have similar and significant effects on Fe when compared to fructose or glucose, though it is not explored is not discussed. Changes in Fe from week 0 to 2 appear to be similar in all groups, including aspartame. A rough calculation suggests a significant change in Fe in the aspartame group.
Calculations indicated that fructose or glucose could only account for about 7% of the change in Cu or Zn, and could not account for the change in Fe, though these diet variables should be the only drivers. Perhaps there is a confounding effect of calorie displacement (though this cannot account for the change in Fe in the aspartame group). Is it possible that the intervention of 25% of calories-worth of beverage might displace food that would otherwise bring some Cu or Zn? If the calories were truly replaced, one could imagine that this is the case. It is also possible that these beverage calories were added and did not displace much, though that data should be available from the study. Perhaps the authors could consider these possibilities.
The Cu-Zn relationship is well documented in literature and should not be surprising, where high Zn can cause Cu deficiency and changes in Cu/Zn ratios have been noted with various pathologies. It is understandable to miss the recent work, as much has been published in 2020. The context of fructose consumption on Cu/Zn relationship is not well documented, but Morrell et al (2020) reported fructose impacting both Cu and Zn in male but not female rats, with fructose impacts on Cu dependent on dietary Cu availability.
In conclusion, this is a great start at intervention studies of the impact of fructose and glucose on Cu and other metals. The careful analysis of ceruloplasmin activity vs. abundance is very useful, as it is often mentioned and rarely executed.
Minor notes below:
Page 2, line 73-74 comment about epidemiological data on trace mineral content in the diet should be supported by a citation.
Page 11, line 282, refers to gender, but it appears the variable is Sex.
The authors might acknowledge that intervention beverages are ~1.5 X concentration of sugar as in typical soft drinks, so the equivalent would be approximately 4.5 12-ounce cans, just to help the relative framing for the reader.
Reviewer 2 Report
Harder et al present a novel and very interesting diet-intervention study correlating the intake of sugar-sweetened beverages with the metabolism of transition metals such as cooper, zinc and iron. Of note, this work indicates that activity of ceruloplasmin ferroxidase activity correlates with serum copper concentration, suggesting that the activity then the levels the enzyme are a better indicator of copper status, with important consequences for the clinical practice. The manuscript is well-written, and the main findings are clearly stated. The authors are also very thorough in indicating potential caveats of the analysis and I would like to congratulate them for the unbiased analysis of their data.
Some aspects could be improved in the presentation/ analysis of the data:
- in table 2, the values with statistical significance should be in bold, as it has been done in the other tables throughout the manuscript.
- in table 4, it should be indicated whether all experimental groups have been used in the analysis. It would be interesting to determine if the glucose and/or fructose independently alter iron and copper concentrations when the subjects were stratified by BMI.
- the number of subjects in each BMI group should be indicated. I also wonder if the BMI of the subjects was calculated after the intervention, and whether potential changes in BMI throughout the study could be masking potential differences in the concentration of iron after the intervention.
In order to deepen the findings of the article, I suggest measuring the levels of hepcidin at baseline and after intervention (stratified by BMI) in order to determine if the changes in iron concentration at the baseline correlate with alterations in hepcidin levels. Along these lines, hepcidin and ferritin are known to be induced by inflammatory cytokines. As the changes in ferritin and transferrin were negatively associated with changes in metabolism markers, the measurement of inflammatory cytokines such as IL-6, INFgamma and TNFalpha (by ELISA or by microbeads array) in the serum of the subjects would allow to understand whether the consumption of sugar-sweetened beverages alter the metabolism of iron due to alterations in the immune system. The authors may also discuss the possible role of the immune system in the observed alterations in metabolism of transition metals.
